# Through bonds or contacts? Mapping protein vibrational energy transfer using non-canonical amino acids

Erhan Deniz [1,6], Luis Valiño-Borau [2,6], Jan G. Löffler [1,6], Katharina B. Eberl[1], Adnan Gulzar[2], Steffen Wolf [2,7], Patrick M. Durkin[3,4], Robert Kaml[3], Nediljko Budisa [3,5,7], Gerhard Stock [2,7✉] & Jens Bredenbeck [1,7✉]

Vibrational energy transfer (VET) is essential for protein function. It is responsible for efficient energy dissipation in reaction sites, and has been linked to pathways of allosteric communication. While it is understood that VET occurs via backbone as well as via non-covalent contacts, little is known about the competition of these two transport channels, which determines the VET pathways. To tackle this problem, we equipped the β-hairpin fold of a tryptophan zipper with pairs of non-canonical amino acids, one serving as a VET injector and one as a VET sensor in a femtosecond pump probe experiment. Accompanying extensive non-equilibrium molecular dynamics simulations combined with a master equation analysis unravel the VET pathways. Our joint experimental/computational endeavor reveals the efficiency of backbone vs. contact transport, showing that even if cutting short backbone stretches of only 3 to 4 amino acids in a protein, hydrogen bonds are the dominant VET pathway.

[1] Institute of Biophysics, Goethe University Frankfurt, Frankfurt/Main, Germany. [2] Biomolecular Dynamics, Institute of Physics, Albert Ludwigs University, Freiburg, Germany. [3] Institute of Chemistry, Technical University Berlin, Berlin, Germany. [4] Present address: GenoSynth GmbH, Magnusstraße 11, Berlin, Germany. [5] Present address: Department of Chemistry, University of Manitoba, Winnipeg, Canada. [6] These authors contributed equally: Erhan Deniz, Luis Valiño-Borau, Jan G. Löffler. [7] These authors jointly supervised this work: Steffen Wolf, Nediljko Budisa, Gerhard Stock, Jens Bredenbeck.
✉email: stock@physik.uni-freiburg.de; bredenbeck@biophysik.uni-frankfurt.de

Vibrational energy transfer (VET) is linked to many important processes in proteins. Chemical reactions in enzymes create substantial excess energy that needs to be efficiently dissipated in order to avoid damage to the protein[1,2]. The most active enzymes can carry out a catalytic cycle producing significant excess energy up to $10^7$ times per second[3]. Light driven reactions for instance, such as the isomerization of chromophores in photoreceptors, can create excess energies in the range of more than 2 eV[4]. Enzymes have been hypothesized to carry out reactions efficiently because the reaction coordinate is coupled to so-called reaction promoting vibrations, that coincide with efficient vibrational energy transfer pathways[5,6]. In the context of allosteric communication, an increasing number of theoretical studies is proposing distinct VET pathways between distant sites to coincide with pathways of allosteric signal propagation[2,7–13].

In the systems and processes mentioned above, VET is believed to proceed efficiently not only along the backbone, but in particular across non-covalent contacts, such as hydrogen bonds, salt bridges, and stacking contacts. However, although there are a number of experimental VET studies covering smaller peptides[14–17] and even proteins[18–22] with quite impressive time resolution, a systematic analysis of backbone versus contact efficiency for VET has not been achieved yet. The short peptides studied before lack a defined secondary or tertiary structure and the associated contacts, hence VET will occur mainly along the backbone. Proteins on the other hand feature a well-defined three-dimensional structure with many non-covalent contacts. However, here the number of potential pathways is becoming very large, so far prohibiting a detailed analysis of pathways in a concerted effort between experiment and theory.

In helical peptides, where VET has already been studied with considerable effort[14,17], non-covalent interactions abound, mainly intrahelical hydrogen bonds. As the residue connection via hydrogen bonds in helices is parallel to the backbone, however, it is difficult to disentangle contributions from backbone and contact transfer. Here, we utilize the very stable β-hairpin structure of a tryptophan zipper (TrpZip2)[23] for a detailed study of backbone and contact contributions to VET (Fig. 1a).

VET occurs on a picosecond timescale and is experimentally studied by femtosecond pump-probe spectroscopy. Vibrational energy is injected into the system either by IR excitation[16,17,24–26] and subsequent relaxation of a local vibration, or by UV/VIS excitation[14,18,20,27] and internal conversion of a chromophore. Energy propagation is monitored by another spectroscopic transition, either vibrational[1,15,17,26,28] or electronic[29].

For VET studies in peptides and proteins we recently introduced a pair of non-canonical amino acids consisting of a vibrational energy donor and a vibrational energy sensor, which can be incorporated either synthetically or cotranslationally during protein expression[16,20]. Enzymatic synthesis of our VET donor azulenylalanine (Azu) has been recently established[30], making it easily accessible. The VET sensor Aha is commercially available due to its main use as a click-chemistry reagent. By varying the position of the VET donor and sensor on the two opposing β-hairpin strands (Fig. 2a), the along-strand and across-strand distance between the donor and sensor can be independently varied, which allows to disentangle backbone and contact contributions.

Accompanying these experimental studies, biomolecular flow of vibrational energy has been described by atomistic molecular dynamics (MD) simulations[7–13,31–37]. Moreover, various network models of energy transport have been proposed[11,35,36], which typically aim to predict the energy flow between specific parts (usually residues) of a protein. In particular, Buchenberg et al.[36]

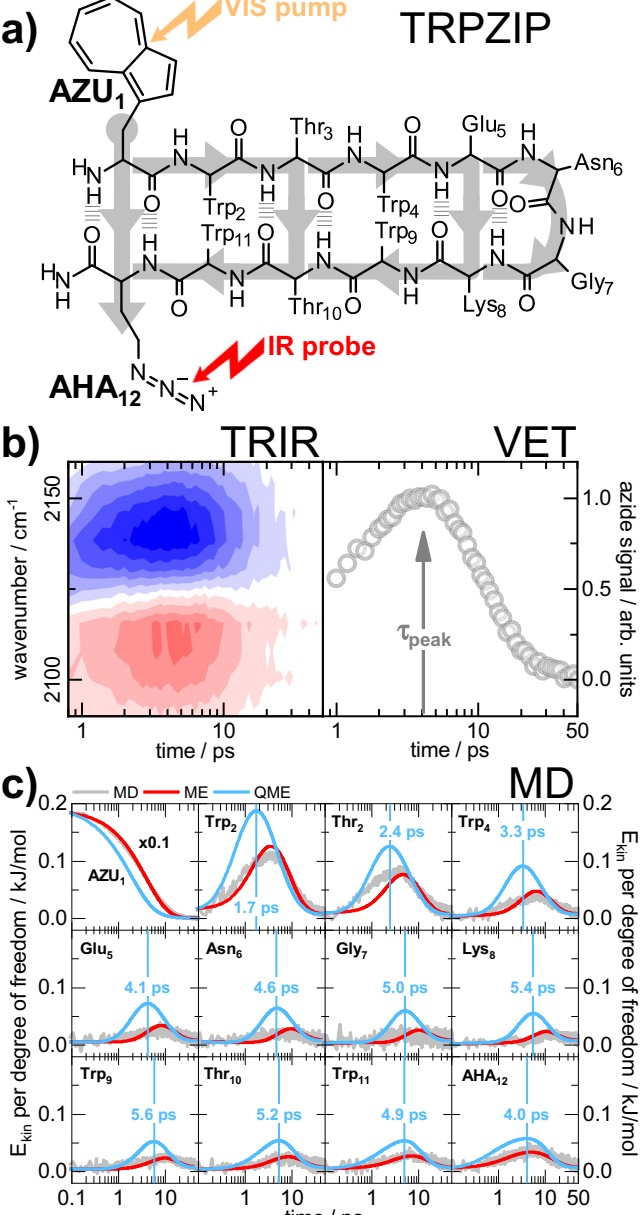

**Fig. 1 Principles of our VET study on TrpZip2. a** Vibrational energy is injected into the system via Azu and propagates along various possible pathways to Aha (gray arrows). **b** (Left): TRIR spectrum of the TrpZip2 variant above (the spectra of all variants are shown in Supplementary Fig. 1). The spectral VET signature consists of an induced (red) and reduced (blue) absorption, respectively. The shape is due to the shift of the azide band to lower wavenumbers upon arrival of vibrational energy. (Right) transient of the total absorption change of the VET signal. Arrow indicates the peak time. **c** Time evolution of residue energies of the variant above obtained from non-equilibrium MD simulations (gray), from the master equation model (ME, red) and from the quantum corrected model (QME, blue). Vertical blue lines indicate the peak times according to the QME model. Source data are provided as a Source Data file.

suggested a master equation

$$\frac{dE_j(t)}{dt} = \sum_i \left[ k_{ij} E_i(t) - k_{ji} E_j(t) \right],$$ (1)

where $E_i$ denotes the kinetic energy of residue $i$ and $k_{ij}$ represents the rate of energy transport from residue $i$ to residue $j$. As the rate

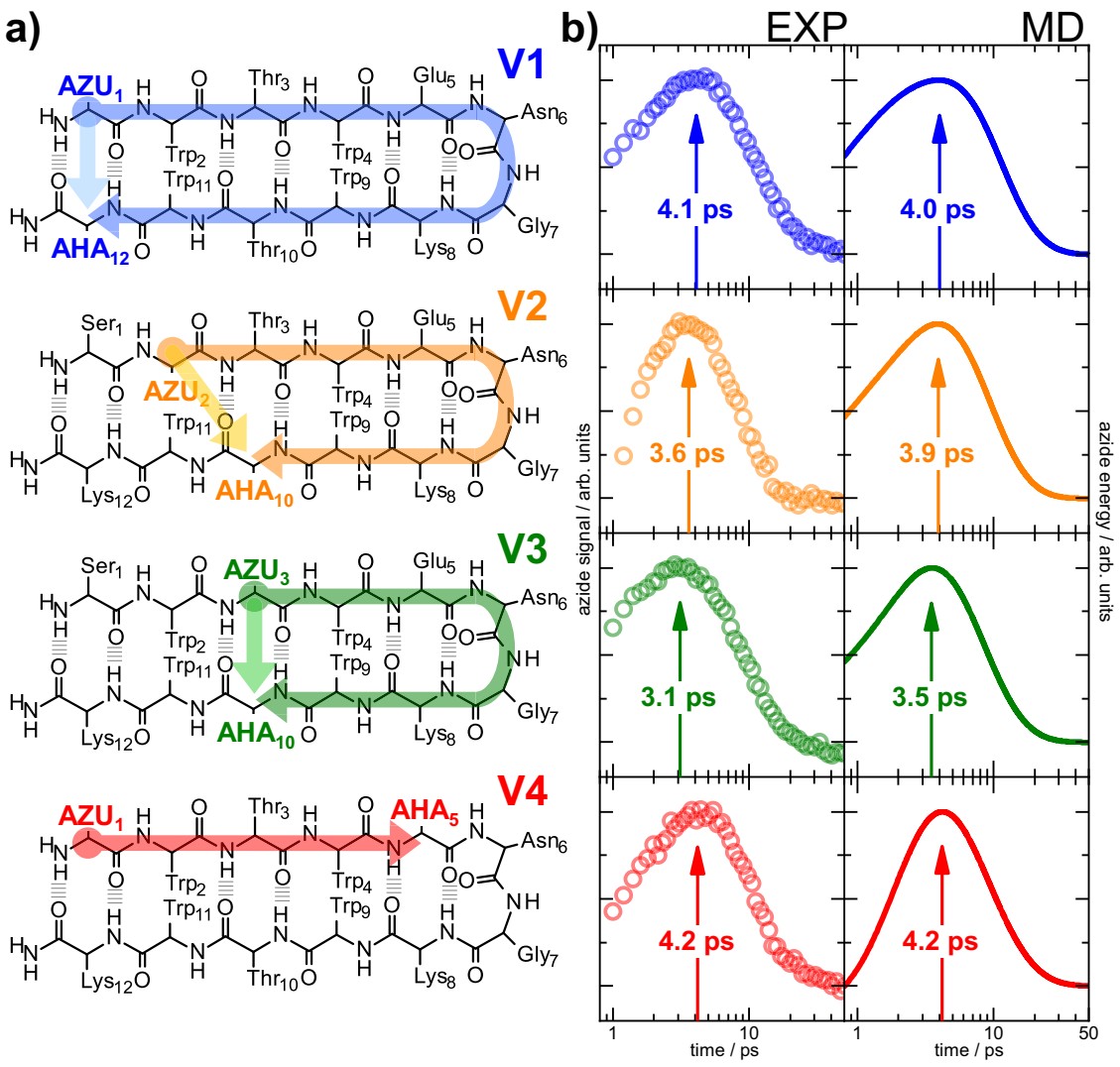

**Fig. 2 Overview of the TrpZip2 variants and their measured and computed VET signals. a** Scheme of our four TrpZip2 variants V1–V4. The systematically varied positions of VET donor-sensor pairs in peptide scaffolds are highlighted with the corresponding color. The shortest pathways along and across strands are marked with colored arrows. **b** VET transients have the same color coding as the TrpZip2 schemes on the left. Arrows indicate the peak times. The panels on the left display the experimentally measured VET transients, while the right panels show the theoretically determined transients using the rate model. Source data are provided as a Source Data file.

matrix $\{k_{ij}\}$ completely determines the time evolution of the system, we can exploit the master equation to obtain the dominant energy transfer pathways in-between two amino acids[37]. Since the rate matrix is obtained from a classical theory, in a final step we introduce quantum corrections of VET[38–40], which are necessary to achieve a quantitative comparison to experiments. Employing extensive non-equilibrium MD simulations[34] combined with a quantum-corrected master equation model[37] we have succeeded here to elucidate the detailed pathways of VET in TrpZip2 model peptides.

## Results
**Design of the tryptophan zippers.** The TrpZip2 variants V1 to V4 (see Fig. 2a) were created by standard Fmoc solid-phase peptide synthesis (see Supplementary Methods for details). The intact hairpin fold was verified by temperature dependent circular dichroism (CD) spectroscopy. It was confirmed that the TrpZips with the VET pair still form the characteristic hairpin structure, providing a stable scaffold for vibrational energy to propagate along well-defined pathways as exemplified in Fig. 1a. Starting

from the vibrational energy donor (Azu), there are many possible pathways towards the sensor (Aha).

**VIS-pump IR-probe spectroscopy.** We performed VIS-pump IR-probe spectroscopy with 150 fs time resolution to monitor VET (see Supplementary Methods for details)[16,20]. A 613 nm pulse pumps the azulene moiety of Azu into its first electronically excited state ($S_1$). Subsequent ultrafast relaxation into the electronic ground state ($S_0$) occurs via internal conversion, converting the absorbed photon energy into vibrational energy in less than a picosecond[41]. This exception to Kasha's rule provides a sharp spatial and temporal starting point for VET.

The azide of Aha responds to increasing vibrational energy at its site by a red-shift of its stretching frequency due to anharmonic coupling to the low-frequency modes populated by VET[24,42,43]. A mid-IR pulse centered around 2120 cm$^{-1}$ probes the azide stretching frequency. This spectral window is free from any other protein signals and thus allows for an accurate background subtraction of only water contributions, resulting in a clean and distinct response of our vibrational energy sensor[20].

VET leads to a characteristic azide signature in the transient IR (TRIR) difference spectrum (Fig. 1b, left). In this VET fingerprint, positive (red) and negative (blue) features represent induced and reduced absorption, respectively. The rise-and-fall in intensity of the azide difference signal in the TRIR spectrum comes from an increasing and decreasing red-shift of the azide band. As the shift is much smaller than the width of the band, the shift cannot be directly quantified but generates a difference signal with an amplitude proportional to the shift. By summing up the absolute values of the time-dependent absorption changes for all pixels carrying the VET signal, we obtain the dynamics of the VET event (Fig. 1b, right). From these transients we extract the time with maximal signal, hereafter called peak time.

**MD simulations**. To aid the interpretation of the above experiments, we performed all-atom explicit-solvent MD simulations of all considered variants of TrpZip2[34]. To mimic the initial heating of Azu via electronic excitation and subsequent ultrafast (~1 ps) internal conversion[41], the resulting vibrational excitation was approximated by an instantaneous temperature jump, where the excess energy $k_B \Delta T$ is chosen to match the $S_0 \rightarrow S_1$ excitation energy of $\approx 2$ eV, resulting in $\Delta T \approx 600$ K. Following the heating of Azu to $T_0 + \Delta T$, we performed 5000 non-equilibrium trajectories of 50 ps length each for all four variants, using the GROMACS package v2016.3[44]. Initial conditions were sampled from 100 ns long equilibrium simulations of TrpZip2[34]. To monitor the vibrational energy flow from the heater through the peptide, we consider the mean kinetic energy per degree of freedom of the $i$th residue, $E_i(t)$, calculated via an ensemble average over all non-equilibrium trajectories (see Supplementary Methods for details).

As an example, Fig. 1c shows the time evolution of the residue energies of V1. Except for the energy of the initially excited heater residue Azu1 that decays rapidly, the residue energies rise on a picosecond timescale to a peak value. If the energy transport occurred predominantly via the peptide's backbone, we would expect a shift of the peak time with increasing sequence distance to the heater. This is the case for the first ~8 residues, before the peak occurs again at earlier times—the first time noticeable for Trp10—resulting from additional energy flow via the β-sheet hydrogen bonds. In particular, we find an increase of the peak energy towards the C-terminus of the peptide, which mainly comes from efficient interstrand H-bond contacts of Aha12 with the heater residue Azu1.

**Master equation analysis**. While non-equilibrium MD simulations provide the time-dependent energy content of all atoms of the considered molecule, this energy flow does not directly reflect single rate constants corresponding, e.g., to the energy transport through a hydrogen bond, because the individual energy fluxes between these atoms cannot directly be disentangled from this data. Hence we construct a master equation model [Eq. (1)] that is fitted to the MD simulation data, in order to determine the full rate matrix of the system. Using two recently established scaling rules, the interresidue rates can be expressed in terms of structural data (such as interresidue distances) and are proportional to two transport coefficients $D_B$ and $D_C$ for backbone and contact transport, respectively, which have been parameterized from MD simulations[36,37] (see Supplementary Methods for details). Moreover, Eq. (1) accounts for the initial energy relaxation of the photoexcited azulene to the vibrations of the backbone atom of Azu via the heater rate $k_h$, and the subsequent dissipation of the vibrational energy into the solvent by the cooling rate $k_s$. Apart from structural information such as interatomic distances, the transport coefficients $D_B$ and $D_C$ and the rates $k_h$ and $k_S$ and potential heater contacts (see below)

are the only free parameters of the model. Comparing the time evolution of the residue energies $E_j(t)$ obtained from MD and master equation, Fig. 1c reveals excellent overall agreement between the two formulations.

To relate time-dependent IR experiments and non-equilibrium MD simulations, we assume that the calculated vibrational excess energy of Aha coincides with the transient red shift of the azide stretch band of Aha that results from coupling to transiently excited low frequency modes[42,43], and hence that experimental and computational peak times can be compared. Aiming at a quantitative comparison, we furthermore need to correct for a significant underestimation of experimental VET rates in classical calculations, in particular if high-frequency vibrations are involved[38,39]. This underestimation results from the neglect of quantum fluctuations such as vibrational zero point energy, which are known to accelerate VET. As a consequence, calculations of VET via classical equilibrium correlation function are often incorrect by several orders of magnitude[39].

**Quantum corrections**. Non-equilibrium calculations, on the other hand, usually start with an initial excess energy that is larger than the zero point energy of the transporting vibrational modes. Hence such calculations require only modest quantum corrections (typically a factor ~ 2–3). Derived from well-established model problems, the magnitude and the origin of the correction is well understood[38,40]. That is, while classical mechanics correctly describes the energy transport between bilinearly coupled harmonic oscillators (regardless of their frequency), it represents a short-time approximation of quantum mechanics for anharmonic (and high-frequency) oscillators. To empirically introduce quantum corrections $Q$ to the above master equation model, it turns out to be sufficient to scale the main transport coefficients obtained from the MD simulations (i.e., the heater rate $k_h$, the backbone diffusion coefficient $D_B$, and the contact transport coefficient $D_C$) such that they reproduce experimental timescales. For example, for the heater rate we make the ansatz $Q = k_h^Q/k_h^C$, where the superscripts $Q$ and $C$ refer to quantum and classical results, respectively. A global fit of the master equation model to the experimental results for all variants of TrpZip2 yields a single quantum correction factor of $Q = 3.1$ for all three transport coefficients (see Supplementary Methods for details).

Energy dissipation to the solvent, on the other hand, happens significantly slower than a typical interresidue VET step, and is therefore only little affected by quantum effects[45]. Indeed, experimental and MD solvent dissipation times $1/k_s$ appear to be in quantitative agreement (on average 6 and 8 ps in experiment and MD, respectively, see Supplementary Fig. 3 and Supplementary Table 1), so we used the experimental values as input for Eq. (1). Figure 1c demonstrates that the quantum-corrected results for the residue energies reach their peak value in about half of the time of the classical calculations. In addition, as intramolecular VET is accelerated while dissipation into the solvent is not, peaks in quantum-corrected residue energies are on average a factor 2 higher than the classical results.

**Experimental VET timings for different donor-sensor positions**. Figure 2b shows the four different TrpZip2 variants we designed for this study. In V1–V3, we positioned the VET donor-sensor pair on opposing strands with varying backbone distances and similar contact distances, as indicated by dark and light-colored arrows, respectively. In V4, however, where the VET pair sits on the same strand, so mainly the backbone distance matters and is highlighted in red. Since VET timing is supposed to correlate with the VET-pair distance, we have

sorted the variants by the length of their backbone pathways, covering a range from 55 Å as the longest for V1 down to 25 Å as the shortest for V4.

In Fig. 2b (right panel) we show the experimentally obtained VET transients of all TrpZip2 variants and their respective peak times. VET occurred on a picosecond timescale with peak times from 3.1 to 4.2 ps. Considering backbone pathways in TrpZip2 covering 4–11 peptide bonds, these short peak times are already quite surprising when compared to VET in the Azu–Aha dipeptide (consisting of just VET donor and sensor, see Supplementary Fig. 2b) peaking at 2.8 ps, but traversing only a single peptide bond. Assuming similar VET rates in TrpZip2 and the dipeptide, the measured VET timings in V1–V3 rather fit to the contact distance between the opposing strands of the hairpin. The rapid transfer already indicates that in the hairpin geometry VET takes shortcuts via bridging non-covalent contacts, moving rapidly from one strand to the other instead of traveling all the way along the backbone.

However, when taking a closer look into the peak times of V1–V3, a trend becomes apparent, which on first glance is in conflict with transport occurring mainly via contacts: Intriguingly, the VET peak times seem to correlate with the backbone distance (indicated with dark-colored arrows in Fig. 2a) of Azu and Aha attached on opposing strands—the shorter the backbone distance, the shorter the peak time. This correlation argues for vibrational energy to cross the β-strands mainly along the backbone.

To test if the observed speed-up from V1 to V3 indeed reports a dominant role of backbone transport, we designed V4, where donor and sensor are located even more closely and on the same strand. In V4 the backbone distance of the donor-sensor pair is even 40% shorter than in V3 which exhibits a peak time of 3.1 ps. Consequently, a peak time well below 3 ps would be expected if backbone transport dominated. In contrast, the peak time of V4 is with 4.2 ps the longest time observed among all variants. Therefore, vibrational energy already needs more than 4 ps to cover only one strand of the β-hairpin. Thus, the shorter peak times in V1–V3 with the donor-sensor pair on the opposite strand can only be explained by an efficient across-strand VET via shortcuts through bridging contacts (indicated with light-colored arrows in Fig. 2a).

**Partial unfolding of the hairpin delays VET**. The important role of transfer between strands via non-covalent contacts is corroborated by measurements of partially unfolded TrpZip2. We repeated our VET measurements in presence of 8 M GdmCl, which is a strongly denaturing condition for the majority of peptides and proteins. Denaturation of TrpZip2 was found to be highly heterogeneous with multiple partially unfolded intermediate states[46], most likely owing to a strong stabilization via hydrophobic contacts between tryptophans that are resistant to denaturing agents. As can be seen in Supplementary Fig. 2a, we managed to unfold V2 and V3 by 25% as monitored by the decrease in signal intensity in the CD spectra. In V3 the partial unfolding distinctively delayed the peak time by 1.0 ps compared to the VET transient of unperturbed hairpin structures. For V2, we observe the same effect to similar extent, shifting the peak time by 0.8 ps. One might expect a somewhat larger delay of the VET in V2 compared to V3 because of the slightly larger through backbone distance. However, due to the heterogeneity of the partially unfolded ensemble, which might also differ between mutants and the fact that different contacts are relevant for VET in V2 and V3, such details in the effect of the small amount of partial unfolding are difficult to predict. Nevertheless, it is clear

that with less intact non-covalent contacts efficient across-strand VET is impeded. Interestingly, the structure of V1 was unperturbed in presence of 8 M GdmCl according to the CD signal, leading to an unperturbed VET transient. We tentatively attribute this stabilization of V1 to the placement of Azu directly at the end of the hairpin motif, where it may hinder the opening of the strand connection via contacts with adjacent tryptophan residues. VET in the dipeptide Azu–Aha is also not affected by high concentration of denaturant, confirming that changes in the energy transfer to the solvent by addition of denaturant are not responsible for the observed changes in VET times (Supplementary Fig. 2b).

**Computational results**. The right panel of Fig. 2b shows the calculated time evolution of the Aha residue energy, as obtained from the quantum master equation introduced above. We find very good agreement between experiment and theory for all four variants: the rise and decay of IR signals and of computed Aha residue energies are very similar, and we find the same order of peak times for the four systems. In line with previous studies on helical peptides[14,15], the agreement indicates that the energy flow in TrpZip2 is diffusive, because this is the underlying assumption of the master equation model. Predominantly ballistic VET in contrast has been observed for molecules with identical repeating units containing less variation in bond strengths, such as alkanes[1,29,47].

Having validated our simulation strategy by comparison to experiment, we are in a position to analyze the resulting master equation model to reveal the preferred pathways of energy flow in TrpZip2 in detail. We consider energy transfer via interstrand hydrogen bonds (HB), backbone vibrations (BB), and heater contacts (HC). The latter are direct van der Waals and electrostatic interactions between the Azu side chain and residues on the opposite strand. While two adjacent residues of the backbone are coupled via a (single) backbone transport rate, two (not covalently bonded) residues can interact via both types of contacts, HB and HC (e.g., residues 1 and 12 in variant 1). Despite the fact that non-polar contacts (e.g., due to the stacking of Trp aromatic rings) are not effective carriers of vibrational energy[36,48], heater contacts need to be considered because of the high energy of the heater[34]. Here, we performed extensive Markov chain Monte Carlo simulations[37] (typically $10^8$ steps for each system) to explore all energy transport pathways between the Azu and Aha residues of TrpZip2.

As an illustration of the preferred class of VET pathways, Fig. 3a shows the time evolution of the fraction of energy that a specific class of pathway contributes to the energy flow from the Azu heater to the Aha probe. (For more details, see Supplementary Fig. 4.) As may be expected, if we excite and probe the same β-strand (as in V4, from Azu1 to Aha5), the energy transport occurs predominantly via the backbone (~85%). On the other hand, if we excite and probe the two ends of TrpZip2 (as in V1, from Azu1 to Aha12), the transport occurs predominantly via inter-strand hydrogen bonds (~68%) and heater contacts (~13%) between these residues, while backbone transport is less important (~19%). The backbone contributions in mutants V2 and V3 amount to 24% and 20%, respectively, while again the major part of the energy is transferred via cross-strand contacts. This is in line with the experimental observation that transfer from one strand to the other is faster than transfer along one strand, and is further supported by the experimental results on partial denaturation experiments, where GdmCl ruptured some of the interstrand hydrogen bonds and hence reduced the contribution from HB transfers.

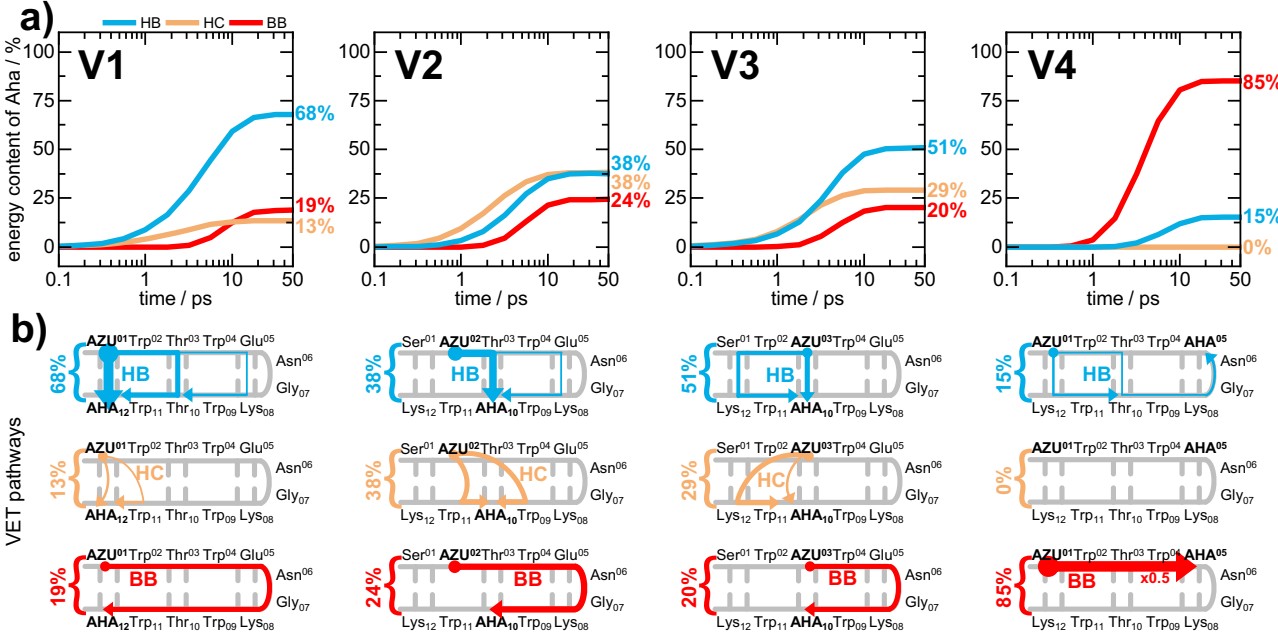

**Fig. 3 Calculated VET pathways from the heater Azu to the sensor Aha in our TrpZip2 variants, obtained from Markov chain Monte Carlo simulations.** **a** Time-dependent contributions to the energy arriving at Aha on pathways including interstrand H-bonds (HB, blue), heater contacts (HC, orange) and solely backbone (BB, red). Numbers indicate the energy contribution of the corresponding class of pathway in percent at 50 ps. **b** Scheme of spatially disentangled VET pathways according to Monte Carlo calculations. Color coding as in the top panel. Arrow thickness scales with the contribution of the corresponding pathway. The VET pair is highlighted with bold capitalized letters. For a better comparability the numbers from the top panel indicating the energy contribution of the corresponding class of pathway are shown. Source data underlying Fig. 3a are provided as a Source Data file.

## Discussion

To identify the pathways of VET in a β-hairpin, we have combined peptide design with non-canonical amino acids and femtosecond IR spectroscopy as well as non-equilibrium MD simulations and master equation modeling. The joint experimental/computational approach has enabled us to quantitatively account for the competition between backbone and contact transport in the peptide. Placing the non-canonical Azu and Aha residues at different positions of the β-hairpin permits to resolve VET in both space and time by injecting and probing vibrational energy site-specifically in a femtosecond IR experiment. On the computational side, our recently proposed master equation model aids the interpretation of the experimental results by revealing the importance of the various pathways of VET (Fig. 3b). Moreover, the comparison of experiment and classical simulations has established a quantum correction factor of ~3 for all intramolecular energy transfer channels, while the intermolecular energy dissipation into the solvent is only little (~20%) affected by quantum effects. Since the microscopic origin of the quantum correction is well understood[38,40], we presume that these findings are transferable to the modeling of VET in other proteins.

As a central result, we have shown that VET between opposite β-strands is dominated by contact pathways. Our study shows that transfer over a hydrogen bond shortcut is about as efficient as transport over a stretch of 3–4 amino acids in the backbone. This is remarkable, since typical backbone transfer times between adjacent residues are 0.2–0.3 ps, while typical contact transfer times are 2–10 ps, i.e., about an order of magnitude slower[37]. The overall VET efficiency of interresidue contacts is a consequence of the diffusive nature of biomolecular VET, whose mean square deviation $\langle x^2(t) \rangle$ scales with time $t$, instead of $t^2$ as in the case of ballistic transport. Hence, less efficient but close-by contact transport outperforms efficient but long-distant backbone transport. Constituting an interresidue force network throughout a protein[49], these observed pathways and their underlying contacts are expected to also be important for allosteric communication.

## Methods

**Sample synthesis.** All peptides were synthesized on a Prelude Peptide Synthesizer by Gyros Protein Technologies (Uppsala, Sweden) using rink amide resin via standard Fmoc solid-phase peptide synthesis employing TBTU and Hünig's base as coupling reagents and DMF as solvent. More details on the synthesis are reported in the Supplementary Methods.

**Sample preparation for laser measurements.** Lyophilized TrpZip2 variants were dissolved in 20 mM phosphate buffer pH 2 to a concentration of approximately 15 mg/ml. For measurements in presence of GdmCl the buffer was prepared with 8 M GdmCl. Samples were loaded into a custom-built mountable CaF$_2$ cell with 100 μm path length.

**VIS-pump IR-probe measurements.** A Ti:Sa regenerative amplifier system (Spitfire, Spectra Physics) with 1 kHz repetition rate generating 800 nm pulses with 100 fs pulse duration and 3 mJ pulse energy pumped two optical parametric amplifiers (OPAs). OPA I generated mid-IR probe pulses centered around 2110 cm$^{-1}$ via difference frequency generation. Pulses were dispersed on a spectrometer (Triax, Jobin Yvon) with a 150 1/mm grating and detected on a 2 × 32 pixels MCT detector (Infrared Associates) cooled with an automated liquid nitrogen refilling system[50]. The detector signal was amplified and integrated with home-built electronics.

OPA II generated 613 nm pump pulses via second harmonic generation in a β-barium-borate (BBO) crystal. The pump pulse energy was 8 μJ. A chopper blocked every other pulse resulting in pump-on/pump-off difference spectra. The polarization between the pump and probe pulse was set to magic angle. The focal beam diameters were 150 and 65 μm FWHM for pump and probe, respectively. A more detailed description of the pump-probe setup is available in the Supplementary Methods.

**Data acquisition and analysis.** Transient IR spectra were recorded with software written in Visual Basic 6 and LabVIEW 2016. Data was analyzed with MatLab R2018a (MathWorks) and OriginPro 2018 (OrginLab). More details on the analysis of the VET signals can be found in the Supplementary Methods.

**MD simulations.** Gulzar et al.[34] recently presented extensive non-equilibrium MD simulations of the energy flow in TrpZip2[23] (PDB entry 1LE1). All MD simulations were performed using GROMACS[44] package v2016.3,

Amber99sb*ILDN forcefield[51–53] and TIP3P water[54]. Parameters for Azu and Aha were reported in refs. [34,55], respectively. Detailed information on system preparation, MD simulation, and vibrational energy monitoring can be found in the Supplementary Methods.

**Master equation**. To describe the flow of vibrational energy in TrpZip2, we employed a master equation model as introduced in refs. [36,37]. Details on the parametrization of this model including applied scaling rules and quantum corrections can be found in the Supplementary Methods.

**Reporting summary**. Further information on research design is available in the Nature Research Reporting Summary linked to this article.

## Data availability
Data supporting the findings of this manuscript are available from the corresponding authors upon reasonable request. A reporting summary for this Article is available as a Supplementary Information file. Source data are provided with this paper. Atomic coordinates of the original TrpZip2 peptide the simulations are based on can be found in the Protein Data Bank under the accession code PDB 1IL1 (see ref. [23]).

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

## Acknowledgements

We acknowledge DFG funding (BR 3746/4-1; BU1404/9-1; INST 161/722-1; STO 247/10-2, INST 39/963-1 FUGG) as well as funding by the Canada Research Chairs Program (grant no. 950-231971). E.D. thanks the Stiftung Polytechnische Gesellschaft for a MainCampus-doctus fellowship. J.B. thanks the Alexander von Humboldt Stiftung for a Sofja Kovalevskaja award. This work has been supported by the DFG via the Research Unit FOR5099 "Reducing complexity of nonequilibrium systems", the High Performance and Cloud Computing Group at the Rechenzentrum of the University of Freiburg, the state of Baden-Württemberg through bwHPC and the DFG (bwForCluster NEMO RV bw18A004), and the Black Forest Grid Initiative.

## Author contributions

N.B., G.S., and J.B. designed research. N.B., S.W., G.S., and J.B. supervised research. E.D., J.G.L., and K.B.E. provided all experimental results. L.V.B. and A.G. provided all computational results. P.M.D. and R.K. provided the synthesized samples. E.D., L.V.B., and J.G.L. contributed equally to this work. All authors wrote the paper.

## Funding

## Competing interests

The authors declare no competing interests.
