## [Peer Review File · Nature Communications]

Reviewer #1 (Remarks to the Author):

The manuscript reports the results of time-resolved spectroscopic measurements and computational studies of vibrational energy transfer (VET) in a beta-hairpin tryptophan zipper (TrpZip2). The study explores competition between energy transfer along the backbone and energy transfer across contacts of TrpZip2 that may not be close in sequence. In the pump-probe experiments non-canonical amino acids are introduced, one that serves as an injector of excess vibrational energy into the molecule and the other as a sensor. The sensors (injector/sensor) were positioned in 4 different ways, combinations of placing them on the same and on different strands and with different numbers of residues between them, so that energy transfer between opposing strands and within the same strand could be studied. The short times (about 4 ps) for VET for the various combinations of sensor placements indicate an important role of non-covalent contacts in this process, particularly upon comparing with one of the systems (V4), for which the through-strand distance between the sensors is relatively short. For V4, no non-covalently bonded contacts are needed for VET between the sensors, and the VET time is longer than for the other systems. Further evidence of the role of non-covalently bonded contacts in VET is provided by denaturation studies, where some of those contacts are lost and the VET times delayed. The computational study involves application of a master equation model, parameterized using results of all-atom simulations, and a quantum correction factor. VET transmission times between the two sensors found in the computational study are in very good agreement with those found experimentally. Close agreement with results of master equation simulations indicates that the VET occurs diffusively, as opposed to ballistically. Pathways for VET between the sensors are identified for the 4 different systems (V1 - V4) from the results of the master equation simulations, detailed in Fig. 3. In this way the authors can determine the fraction of VET between sensors that occurs via backbone contributions, via inter-strand hydrogen bonds and via heater contacts.

VET is an important process in protein function and the combined experimental-computational study provides novel insights into details of this process. The manuscript will interest many readers of Nature Communications and publication is recommended after the authors consider the following:

1. The results of the study of denatured TrpZip2 are interesting. V1 is apparently not denatured and the VET time remains unchanged, whereas V2 and V3 are partially denatured, and the VET time increases by 0.8 ps for V2 and 1.0 ps for V3. Naively, one might have expected a larger delay for V2, since the alternative pathway, upon breaking non-covalent contacts, is a longer traverse through the strand for V2 compared to V3. The authors might comment on this.
2. The authors obtain an empirical quantum correction factor (3.1) from fitting computational results using the master equation model to the VET measurements of the 4 TrpZip2 systems that were studied, Do they expect that this value is transferrable to model VET across contacts in other protein systems, at least the kinds of contacts through which energy transfer was modeled in V1 - V4?
3. The rate constant for energy transfer to the solvent computed from the MD simulations compares well with the results inferred from the experiments and thus does not require a correction factor. Do the authors expect this to be generally the case, or would it generally depend on the solvent used? Incidentally, the solvent is apparently water, though that is not stated in the manuscript, only in the SI. It might be helpful to point out that the solvent is in fact water somewhere in the manuscript.

Reviewer #2 (Remarks to the Author):

This is a very interesting article which analyzes the mechanism of vibrational energy transfer (VET) in a TRPZIP mini-protein. The strengths of the work are: combination of experiments with modeling, thoughtful design of the model systems for study and a thorough theoretical analysis of the energy flow pathways. The experiments and molecular dynamics simulations are well designed and carefully analyzed. The signal modeling through a Master Equation approach is quite insightful. The findings of VET occurring both through backbone and inter-residue contacts are novel and highly interesting. This work should be of interest to a wide range of readers. Prior to publication, I would like the authors to respond to these comments:

1. The modified TRPZIP systems include non-standard residues. It would be helpful if the authors provided some information on the parameterization of these residues - e.g a reference or SI entry.
2. It is satisfying to see that nonequilibrium MD simulations can predict rates of VET that are within a factor of two of the experimental findings. However, it is disappointing that the MD results do not provide enough information for determining all the rate constants in the Master Equation. Could some information about the VET paths be obtained from the energy flow patterns seen in the MD, to complement the stochastic modeling?
3. It might be helpful for readers if some more information was provided about the kinetic modeling. How many free parameters are in the Master Equation models? Also, how is the classification made into HB, HC and BB classes? Is each pair assigned to one class or are there separate contributions to the VET rates k_{ij} from the different mechanisms? E.g. are there residue pairs that are both neighbors in sequence (BB) and involved in contacts (HC), or both HC and HB?

Reviewer #3 (Remarks to the Author):

In the manuscript, the authors aimed at addressing an interesting question about the vibrational energy relaxation in protein; pathway through backbone vs. pathway through hydrogen bond. To study this, the authors chose a system of beta-hairpin fold and pairs of non-canonical amino acids. The data and analysis were well done both experimentally and theoretically, and I enjoy reading the manuscript. On the other hand, the authors should have presented more detailed experimental data and explanation on the simulation, in my opinion. The details are given below.

1. TRIR spectra (like Figure 1b) corresponding to Figure 2 should be presented in SI. The authors can (and probably should) discuss the frequency shift of the peak frequency as a function of time.
2. In the SI, the simulation details should be written (including more detailed information in SI than in main text). In particular, I do not understand how the authors fit the master equation to the experimental data. "Details of the fitting procedure are described elsewhere" without any citations does not help understanding of readers.

3. Related with 2, it is a bit questionable for me that the different time scale of the simulation and experiment were attributed to the quantum effects of the vibrational energy flow. The importance of the quantum treatment has been emphasized by Skinner as well as the author, while it is known that the anharmonic coupling of the vibrational modes is also essential for vibrational energy relaxation process of, for example, water. If the anharmonic coupling is important, the use of the force field model including only the harmonic potential (Amber) seems inadequate for monitoring the vibrational energy transfer. A short discussion or outlook that can address this question should be added.

Legend:

italics (tabbed): our reply

bold+underlined: modifications to the manuscript

‘single quotes’: text from the manuscript

Reviewer #1 (Remarks to the author):

The manuscript reports the results of time-resolved spectroscopic measurements and computational studies of vibrational energy transfer (VET) in a beta-hairpin tryptophan zipper (TrpZip2). The study explores competition between energy transfer along the backbone and energy transfer across contacts of TrpZip2 that may not be close in sequence. In the pump-probe experiments non-canonical amino acids are introduced, one that serves as an injector of excess vibrational energy into the molecule and the other as a sensor. The sensors (injector/sensor) were positioned in 4 different ways, combinations of placing them on the same and on different strands and with different numbers of residues between them, so that energy transfer between opposing strands and within the same strand could be studied. The short times (about 4 ps) for VET for the various combinations of sensor placements indicate an important role of non-covalent contacts in this process, particularly upon comparing with one of the systems (V4), for which the through-strand distance between the sensors is relatively short. For V4, no non-covalently bonded contacts are needed for VET between the sensors, and the VET time is longer than for the other systems. Further evidence of the role of non-covalently bonded contacts in VET is provided by denaturation studies, where some of those contacts are lost and the VET times delayed. The computational study involves application of a master equation model, parameterized using results of all-atom simulations, and a quantum correction factor. VET transmission times between the two sensors found in the computational study are in very good agreement with those found experimentally. Close agreement with results of master equation simulations indicates that the VET occurs diffusively, as opposed to ballistically. Pathways for VET between the sensors are identified for the 4 different systems (V1-V4) from the results of the master equation simulations, detailed in Fig. 3. In this way the authors can determine the fraction of VET between sensors that occurs via backbone contributions, via inter-strand hydrogen bonds and via heater contacts.

VET is an important process in protein function and the combined experimental-computational study provides novel insights into details of this process. The manuscript will interest many readers of Nature Communications and publication is recommended after the authors consider the following:

> *We thank the reviewer for his/her comments and the positive evaluation.*

(1) The results of the study of denatured TrpZip2 are interesting. V1 is apparently not denatured and the VET time remains unchanged, whereas V2 and V3 are partially denatured, and the VET time increases by 0.8 ps for V2 and 1.0 ps for V3. Naively, one might have expected a larger delay for V2, since the alternative pathway, upon breaking non-covalent contacts, is a longer traverse through the strand for V2 compared to V3. The authors might comment on this.

> Indeed, a slightly larger effect of unfolding on the VET time in V2 compared to V3 would be expected if unfolding was a two state process, changing the equilibrium between a fully folded state (all contacts present) and a fully unfolded state (no contacts, only backbone transfer possible). However, it is known [Ref. 46 and references therein] that unfolding of TrpZip2 is not a two-state process but involves a distribution of partially folded states. Furthermore, different contacts play a role for VET in V2 as compared to V3, as can be seen in Figure 3b). A simple prediction if the similar small increase in partially unfolded structures achieved under our unfolding conditions causes a larger increase in VET time in V2 or V3 is therefore not possible.

To make this point clearer in the main text, we now write towards the end of the Section "Partial unfolding of the hairpin delays VET":

‘In V3 the partial unfolding distinctively delayed the peak time by 1.0 ps compared to the VET transient of unperturbed hairpin structures. For V2, we observe the same effect to similar extent, shifting the peak time by 0.8 ps. **One might expect a somewhat larger delay of the VET in V2 compared to V3 because of the slightly larger through backbone distance. However, due to the heterogeneity of the partially unfolded ensemble which might also differ between mutants and the fact that different contacts are relevant for VET in V2 and V3, such details in the effect of the small amount of partial unfolding are difficult to predict. Nevertheless, it is clear that with less intact non-covalent contacts efficient across-strand VET is impeded.** Interestingly, the structure of V1 was unperturbed in presence of 8 M GdmCl according to the CD signal, leading to an unperturbed VET transient.’

(2) The authors obtain an empirical quantum correction factor (3.1) from fitting computational results using the master equation model to the VET measurements of the 4 TrpZip2 systems that were studied, do they expect that this value is transferrable to model VET across contacts in other protein systems, at least the kinds of contacts through which energy transfer was modeled in V1-V4?

> Yes we do. While the comparison to experiment presented in this work can provide only empirical evidence, it has been shown for various well-understood model problems (Refs. 38, 40) that the quantum correction factor is in the range of 2–3 for vibrational frequencies typically found in protein energy transport.

To emphasize this point, we now write at the beginning of the section "Quantum corrections" in the main text.:

‘Hence such calculations require only modest quantum corrections, typically a factor $\sim 2-3$. **Derived from well-established model problems, the magnitude and the origin of the correction is well understood.**^{38,40} That is, while classical mechanics correctly describes the energy transport between bilinearly coupled harmonic oscillators (regardless of their frequency), it represents a short-time approximation of quantum mechanics for anharmonic (and high-frequency) oscillators. To empirically introduce quantum corrections Q to the above master equation model, ...’

as well as at the end of the first paragraph of the Conclusions:

‘Moreover, the comparison of experiment and classical simulations has established a quantum correction factor of ~3 for all intramolecular energy transfer channels, while the intermolecular energy dissipation into the solvent is only little (~20%) affected by quantum effects. **Since the microscopic origin of the quantum correction is well understood^{38,40}, we presume that these findings are transferable to the modeling of VET in other proteins.**’

(3.1) The rate constant for energy transfer to the solvent computed from the MD simulations compares well with the results inferred from the experiments and thus does not require a correction factor. Do the authors expect this to be generally the case, or would it generally depend on the solvent used?

> In general, the energy dissipation to the solvent is about an order of magnitude slower than a typical inter-residue VET step. This indicates that mainly low-frequency modes are involved, which are only little affected by quantum effects. Since water is among the strongest coupled solvents, we expect that the energy dissipation to the solvent can be generally well described by a classical formulation. On the other hand, the energy dissipation may depend sensitively on the force field model of the solvent. Since these models may be of different quality for different solvents, the accuracy of the theoretical description can in practice depend on the solvent.

(3.2) Incidentally, the solvent is apparently water, though that is not stated in the manuscript, only in the SI. It might be helpful to point out that the solvent is in fact water somewhere in the manuscript.

> We thank the reviewer for pointing out that this important piece of information is actually missing in the main text.

We added this information in the “VIS-pump-IR-probe spectroscopy” section:

‘This spectral window is free from any other protein signals and thus allows for an accurate background subtraction of only **solvent water** contributions, resulting in a clean and distinct response of our vibrational energy sensor.²⁰’

Reviewer #2 (Remarks to the author):

This is a very interesting article which analyzes the mechanism of vibrational energy transfer (VET) in a TRPZIP mini-protein. The strengths of the work are: combination of experiments with modeling, thoughtful design of the model systems for study and a thorough theoretical analysis of the energy flow pathways. The experiments and molecular dynamics simulations are well designed and carefully analyzed. The signal modeling through a Master Equation approach is quite insightful. The findings of VET occurring both through backbone and inter-residue contacts are novel and highly interesting. This work should be of interest to a wide range of readers.

> We thank the reviewer for his/her comments and the positive evaluation.

Prior to publication, I would like the authors to respond to these comments:

(1) The modified TRPZIP systems include non-standard residues. It would be helpful if the authors provided some information on the parameterization of these residues - e.g., a reference or SI entry.

> We agree with the reviewer, that information on the parametrization of our non-canonical residues Aha and Azu would be helpful.

The information is now given at the beginning of the Section "MD simulations" of the SI:

*'All MD simulations were performed using GROMACS³ package v2016.3, Amber99sb*ILDN forcefield⁴⁻⁶ and TIP3P water.⁷ **Parameters for Azu and Aha were reported in Ref. 1 and Ref. 8, respectively.** Na⁺ and Cl⁻ were added at a salt concentration of 0.1 M ...*

(2) It is satisfying to see that nonequilibrium MD simulations can predict rates of VET that are within a factor of two of the experimental findings. However, it is disappointing that the MD results do not provide enough information for determining all the rate constants in the Master Equation. Could some information about the VET paths be obtained from the energy flow patterns seen in the MD, to complement the stochastic modeling?

> We agree with the reviewer that it is at first sight somewhat surprising that all-atom simulations would not provide this information directly.

We now explain this at the beginning of the Section "Master equation analysis":

'While non-equilibrium MD simulations provide the time-dependent energy content of all atoms of the considered molecule, this energy flow does not directly reflect single rate constants corresponding, e.g., to the energy transport through a hydrogen bond, because the individual energy fluxes between these atoms cannot directly be disentangled from this data. Hence we construct a master equation model [Eq. (1)] that is fitted to the MD simulation data, in order to determine the full rate matrix of the system. In a second step, a master equation model Eq. (1) is constructed. Using two recently established scaling rules, ...'

(3) It might be helpful for readers if some more information was provided about the kinetic modeling. How many free parameters are in the Master Equation models? Also, how is the classification made into HB, HC and BB classes? Is each pair assigned to one class or are there separate contributions to the VET rates k_{ij} from the different mechanisms? E.g. are there residue pairs that are both neighbors in sequence (BB) and involved in contacts (HC), or both HC and HB?

> We thank the reviewer for his/her suggestion to include this information on the kinetic modeling.

To address the first point, we added towards the end of the Section "Master equation analysis":

'... and the subsequent dissipation of the vibrational energy into the solvent by the "cooling rate" k_S . Apart from structural information such as inter-atom distances, the transport coefficients D_B and D_C and

the rates k_h and k_s and potential heater contacts (see below) are the only free parameters of the model. Comparing the time evolution of the residue energies ...?

To address the second point, in addition to the Figure 3b, which separates the energy-flow channels backbone (BB), hydrogen bond (HB) and heater contact (HC) of each TrpZip2 variant, we now added to the next-to-last paragraph of the Results:

‘The latter are direct van der Waals and electrostatic interactions between the Azu side chain and residues on the opposite strand. **While two adjacent residues of the backbone are coupled via a (single) backbone transport rate, two (not covalently bonded) residues can interact via both types of contacts, HB and HC (e.g., residues 1 and 12 in variant 1).**’

Reviewer #3 (Remarks to the author):

In the manuscript, the authors aimed at addressing an interesting question about the vibrational energy relaxation in protein; pathway through backbone vs. pathway through hydrogen bond. To study this, the authors chose a system of beta-hairpin fold and pairs of non-canonical amino acids. The data and analysis were well done both experimentally and theoretically, and I enjoy reading the manuscript.

> *We thank the reviewer for his/her helpful comments and the positive evaluation.*

On the other hand, the authors should have presented more detailed experimental data and explanation on the simulation, in my opinion. The details are given below.

(1.1) TRIR spectra (like Figure 1b) corresponding to Figure 2 should be presented in SI.

> *To address this point, **we included an additional figure (Fig. S1) containing the TRIR spectra of all four trpzip2 variants in the SI.** All following Sections and Figures in the SI were renumbered accordingly and a reference to Fig. S1 was added in the caption of Fig.1.*

Figure S1. Overview of TRIR spectra (left) and VET transients (right) of all four trpzip2 variants, analogous to Figure 1b.

(1.2) The authors can (and probably should) discuss the frequency shift of the peak frequency as a function of time.

> As mentioned in the manuscript the VET signal of the azide is generated by ‘increasing vibrational energy at its site by a red-shift of its stretching frequency due to anharmonic coupling to the low-frequency modes populated by VET.’ The shift, however, is very small compared to the linewidth of the transition and can thus not be determined directly. We estimate it to be in the range of 0.2 cm^{-1} where the signal is maximal, two orders of magnitude smaller than the width of the absorption band. Hence, we are well in the limit where the amplitude of the difference spectrum between the induced absorption band and the bleach is proportional to the shift. The time dependence of the shift thus is in a very good approximation the same as the time dependence of the signal amplitude which we discuss.

We added a note regarding the time-dependent frequency shift of the azide band to the last paragraph of the Section “VIS-pump-IR-probe spectroscopy” in our Experimental Strategy:

‘In this VET fingerprint, positive (red) and negative (blue) features represent induced and reduced absorption, respectively. **The rise-and-fall in intensity of the azide difference signal in the TRIR spectrum comes from an increasing and decreasing red-shift of the azide band. As the shift is much smaller than the width of the band, the shift cannot directly quantified but generates a difference signal with an amplitude proportional to the shift.** By summing up the absolute values of the time-dependent absorption changes for all pixels carrying the VET signal, we obtain the dynamics of the VET event (Figure 1b, right).’

(2) In the SI, the simulation details should be written (including more detailed information in SI than in main text). In particular, I do not understand how the authors fit the master equation to the experimental data. "Details of the fitting procedure are described elsewhere" without any citations does not help understanding of readers.

> *We thank the reviewer to point out that more details on the simulation and our fitting procedure are necessary for the readers to understand our computational strategy.*

Additional simulation details are now given at the beginning of the Section "MD simulations" of the SI.

‘Gulzar et al.¹ recently presented extensive non-equilibrium MD simulations of the energy flow in TrpZip2² (PDB entry 1LE1). All MD simulations were performed using GROMACS3 package v2016.3, Amber99sb*ILDN forcefield⁴⁻⁶ and TIP3P water.⁷ **Parameters for Azu and Aha parameters were reported in Ref. 1 and Ref. 8, respectively. Na⁺ and Cl⁻ were added at a salt concentration of 0.1 M, with an excess of Cl⁻ to compensate the net positive charge (+2) of TrpZip2. Long-range electrostatic interactions (distances > 1.2 nm) were computed by the Particle Mesh Ewald (PME) method,⁹ short-range electrostatic interactions were treated explicitly using a Verlet cut-off scheme. After energy minimization, a 10 ns NPT equilibration run was performed. In all equilibrium simulations, we used an integration time step of 2 fs and maintained a temperature of 300 K (via the Bussi thermostat,¹⁰ coupling time 0.1 ps) and a pressure of 1 bar (via the Berendsen barostat,¹¹ coupling time 0.1 ps).** Following suitable equilibrium runs for each TrpZip2 variant considered...’

Regarding the fit of the master equation to the experimental data, we now write in the Section "Quantum corrections":

‘To empirically introduce quantum corrections Q to the above master equation model, it turns out to be sufficient to scale the main transport coefficients **obtained from the MD simulations** (i.e., the heater rate k_h , the backbone diffusion coefficient D_B , and the contact transport coefficient D_C) such that they reproduce experimental timescales.’

as well as in the second paragraph of the "Quantum corrections" Section in the SI:

‘In this work, an empirical quantum correction was determined by fitting master equation results to experimental peak times. **To this end, we applied the quantum correction factor Q to the coefficients D_B and D_C for backbone and contact transport as well as the heating rate k_h and heater contact rates, and performed a global fit of Q such that the quantum-corrected master equation results reproduces**

best the experimental peak times. In this way we obtained a quantum correction factor of 3.1, ~~which was applied to the master equation parameters DB for backbone transport and DC for contact transport, as well as the heating rate $1/kh$.~~ Details of the fitting procedure are described ~~elsewhere, in Ref. 17.~~

(3) Related with 2, it is a bit questionable for me that the different time scale of the simulation and experiment were attributed to the quantum effects of the vibrational energy flow. The importance of the quantum treatment has been emphasized by Skinner as well as the author, while it is known that the anharmonic coupling of the vibrational modes is also essential for vibrational energy relaxation process of, for example, water. If the anharmonic coupling is important, the use of the force field model including only the harmonic potential (Amber) seems inadequate for monitoring the vibrational energy transfer. A short discussion or outlook that can address this question should be added.

> We agree with the Reviewer that anharmonic couplings are essential for vibrational energy relaxation. However, we want to point out that biomolecular force fields do contain many nonlinear (i.e., anharmonic) terms, e.g., the protein interacts with itself and with the surrounding water via van der Waals and Coulomb interactions. These interactions also affect the forces acting on bond and angle vibrations.

Moreover there is a subtle connection between the anharmonicity and the quantum behavior of a system. That is, classical mechanics correctly describes the energy transport between bilinearly coupled harmonic oscillators (regardless of their frequency), see Ref. [38]. As a consequence, quantum effects are important mainly for anharmonic (and high-frequency) oscillators, which in turn explains (at least in part) why anharmonic couplings are essential for VET.

To better explain this issue, we rewrote the first part of the Section 'Quantum correction'

*'Non-equilibrium calculations, on the other hand, usually start with an initial excess energy that is larger than the zero point energy of the transporting vibrational modes, ~~and can be considered as a short time approximation of quantum mechanics.~~ Hence such calculations require only modest quantum corrections, typically a factor $\sim 2-3$, ~~whose microscopic origin is well understood from first principles.~~³⁸ **Derived from well-established model problems, the magnitude and the origin of the correction is well understood.**^{38,40} **That is, while classical mechanics correctly describes the energy transport between bilinearly coupled harmonic oscillators (regardless of their frequency), it represents a short-time approximation of quantum mechanics for anharmonic (and high-frequency) oscillators.** To empirically introduce quantum corrections Q to the above master equation model, ...'*

Reviewer #1 (Remarks to the Author):

The authors have addressed all of my concerns with the original manuscript in their revision. Publication is now recommended.

Reviewer #2 (Remarks to the Author):

The authors have provided satisfactory responses to all comments. I recommend publication of the revised manuscript.

Reviewer #3 (Remarks to the Author):

The manuscripts are now self-consistent and the discussion is robust. Based on my judgement, I would like to recommend the editor to accept this manuscript for publication of Nature Comm. This is a great work.